# Congenital Muscular Torticollis—Current Understanding and Perinatal Risk Factors: A Retrospective Analysis

**DOI:** 10.3390/healthcare12010013

**Published:** 2023-12-20

**Authors:** Janusz Płomiński, Jolanta Olesińska, Anna Malwina Kamelska-Sadowska, Jacek Józef Nowakowski, Katarzyna Zaborowska-Sapeta

**Affiliations:** 1Prof. Adam Gruca Independent Public Clinical Hospital CMKP, 05-400 Otwock, Poland; plominsky@poczta.onet.pl; 2Centre of Postgraduate Medical Education, 01-813 Warsaw, Poland; 3Department of Physiotherapy, College of Rehabilitation, 01-234 Warsaw, Poland; jolantaolesinska6@gmail.com; 4Department of Physiotherapy, Prof. Jan Bogdanowicz Children’s Hospital, 03-924 Warsaw, Poland; 5Department of Rehabilitation and Orthopedics, School of Medicine, Collegium Medicum, University of Warmia and Mazury in Olsztyn, 10-719 Olsztyn, Poland; rinkamelska@gmail.com; 6Regional Specialised Children’s Hospital, 10-561 Olsztyn, Poland; 7Department of Ecology and Environmental Protection, University of Warmia and Mazury in Olsztyn, 10-719 Olsztyn, Poland; jacek.nowakowski@uwm.edu.pl

**Keywords:** torticollis, sternocleidomastoid muscle, children, perinatal risk factors

## Abstract

Introduction: Congenital muscular torticollis (CMT) is an asymmetrical head position resulting from structural changes in the sternocleidomastoid (SCM) muscle that occurs early during a child’s development or due to perinatal trauma. Children with CMT exhibit a marked imbalance in tension between the SCMs. In a typical clinical picture, an ultrasound scan is performed to reveal characteristic lesions, such as tissue fibrosis or post-traumatic changes. An early diagnosis of CMT in newborns and the implementation of treatment offer the chance of a complete resolution. Torticollis treatment aims to restore the SCM’s normal function. Surgical treatment is performed when conservative methods fail to improve the patient’s condition. The indications that surgery is needed include a marked shortening of the SCM, persistent fibrosis in the muscle, constant head and facial asymmetry, and rotation or lateral flexion in the cervical spine restricted by >15°. Of all the newborn and infant anomalies, congenital torticollis is the third most common after hip dysplasia and equinovarus deformities. Some authors demonstrate that torticollis coexists with hip dysplasia. Aim: The aim of this study was to collect data on infants referred to paediatric rehabilitation and to identify the risk factors associated with CMT in this group of patients, as well as to assess demographic and clinical characteristics concerning risk factors. Materials and methods: The target population for this retrospective study consisted of 111 infants aged 0 to 5 months born in Poland and diagnosed with and undergoing treatment due to CMT. The following were determined: the relationship between the side of the CMT location and the type of delivery (caesarean section vs. vaginal), the relationship between the body weight at birth and the side of the CMT location, the relationship between the extent of SCM thickening and the type of delivery, and the incidence of CMT depending on the order of delivery. Results and conclusions: The data revealed that CMT is less common in female infants (*n* = 51, 46%) compared to male (*n* = 61, 54%) infants, in whom a greater birth weight was reported (*p* < 005). Seventy-six percent (76%) of the paediatric patients with CMT were the offspring of primipara mothers. More often, children born via vaginal delivery had left-sided torticollis with a more significant broadening of the SCM, as shown on ultrasound scans, than right-sided torticollis. Theories of torticollis development pathophysiology should be deepened and systematised, and further research is needed.

## 1. Introduction

Congenital muscular torticollis (CMT) is a condition commonly diagnosed at or soon after birth, characterised by an involuntary and asymmetrical head position caused by the unilateral shortening of the sternocleidomastoid (SCM) muscle, usually associated with fibrotic mass [1]. The shortening of the muscle can be detected on an ultrasound scan, where it is seen as a typical morphological lesion (muscle tissue fibrosis). It can occur during prenatal development or due to perinatal trauma. Fibrotic changes result in SCM shortening and secondary mobility restriction in the cervical region of the spine. Figure 1 shows the infant’s characteristic tilt of the head, a rotation to the opposite side of the affected muscle, and the increased thickness of the affected muscle. Two types of torticollis are distinguished, one resulting from intrauterine foetal malposition and the other following perinatal trauma accompanied by a haematoma within the SCM [2]. Bilateral contracture of the SCM muscle is a very rare form of muscle skeletal disorder [3,4]. Of all the newborn and infant anomalies, congenital torticollis is the third most common after hip dysplasia and equinovarus deformities. Some authors demonstrate that torticollis coexists with hip dysplasia [5,6]. An early diagnosis of CMT in newborns and infants and the implementation of treatment offer a chance of complete resolution [7]. The commencement of physiotherapy treatment in the first few months of age contributes to the rapid regression of torticollis symptoms. CMT treatment can be conservative or surgical, depending on its severity. A delayed diagnosis is accompanied by asymmetry of the head and cervical spine region. Asymmetrical loads on the hip and pelvic joints can result in remote deformities. When left untreated, CMT may result in a progressive limitation of head movement, which may end up in eye movement disorders, craniofacial asymmetry, malocclusion, visual defects, neck pain and a compensatory asymmetrical spine curvature that worsens with age [8].

### 1.1. Aetiology

Usually, torticollis is not a diagnosis, but rather a manifestation of a variety of underlying conditions. However, in general, torticollis is classified as either a congenital (present at birth) or acquired (occurring later in infancy or childhood) disorder [1], and the congenital type is further classified into postural, muscular and SCM mass [9]. 

The literature most often cites the theory of muscle injury and remodelling during the foetal period or the theory of an injury during delivery [1].

Concerning the clinical features, torticollis could be divided into two groups of mechanisms.

#### 1.1.1. CMT Type I

Type I CMT is caused by foetal malposition [10]. During pregnancy, the foetal head and neck position can result in selective damage to one of the SCMs. Damage mechanisms include local crushing and ischaemia, which lead to degenerative changes. Factors contributing to foetal malposition may include limited intrauterine space or low amniotic fluid levels. CMT is often observed in infants in a breech position and in primiparas, especially in cases of insufficient amniotic fluid and mechanical pressure of the uterine walls on the foetus [11]. Tissue ischaemia results in muscle oedema and subsequent degenerative changes, ultimately leading to muscle fibrosis. 

#### 1.1.2. CMT Type II

CMT type II is a torticollis resulting from perinatal injury. This occurs most commonly during breech delivery or when there is a disproportion between the size of the foetus and the birth canal. Some authors believe that an abnormal head tilt may cause injury to the mother’s womb. The tilted head prevents proper entry into the birth canal [12]. Other authors claim that the injury to the SCM muscle may be due to excessive twisting of the head during its evacuation when the muscle fibres are torn, and a haematoma is formed, which subsequently undergoes resorption and fibrosis [13]. Others support the trauma theory and argue that traumatic and ischaemic factors act simultaneously on the SCM muscle during labour. The muscle is likely stretched and locally crushed at birth, eventually leading to fibrosis. It has been shown that the affected SCM in patients with late-stage CMT (age ≥1 year) presents no significant blood flow signal, which may be associated with the organisation of the granulation tissue, increased fibrosis, and a reduced number of blood vessels. The disturbances in blood flow or even no blood flow signal within the lesion could indicate significant SCM fibrosis. In these cases, surgery rather than rehabilitative therapy is mandatory [14].

Most articles have not yet provided conclusive evidence, and the full mechanism of CMT is still unknown.

### 1.2. Epidemiology

CMT is a common postural deformity affecting 0.3% to 16.0% of newborns in European countries [15,16,17,18], whereas in Asian countries, the prevalence of CMT ranges from 0.0084% to 3.92% [19,20]. According to Noga, congenital muscular torticollis is diagnosed in 3–5 out of 1000 newborn babies [21]. At the rehabilitation department of the Prof. Jan Bogdanowicz Children’s Hospital in Warsaw, Poland, infants with CMT accounted for 4.7% of all children aged 0–3 years receiving care in the years 2000 to 2018. 

### 1.3. Clinical Presentation

An asymmetrical head position is characteristic of infants with CMT. Typically, the head is tilted laterally on the side of the shortened muscle and rotated with a slight extension towards the opposite side. In some cases, the flattening of the newborn’s skull is noticeable in the parieto-occipital region. This deformity is caused by pressure on the lambdoid suture at the end of pregnancy. Also, occipital plagiocephaly causes a flattening of one side of the back of the head and is often a result of the infant consistently lying on his or her back. Asymmetry within the trunk region is also observed. A child positions the trunk in a “C” shape, with the pelvis positioned obliquely (Figure 1 and Figure 2).

Children with CMT exhibit a marked imbalance in tension between the SCM muscles. On the torticollis side, the tension is excessive, whereas on the opposite side, the muscle fibres are stretched and weakened. In the case of CMT, the thickening and tension of the SCM muscle become palpable as early as between the second and third weeks of age. Thickening is usually observed in the central, oval-shaped muscle. The palpable thickening of the SCM belly is firm, spindle-shaped, painless, and covered with healthy skin. It is most frequently located in the middle or lower third of the CMT (Figure 3). Thickening reaches its maximum size at three or four weeks of age [21]. In CMT, the range of cervical spinal mobility is typically normal. When secondary contracture of the SCM muscle occurs, an examination reveals rotation restriction in the direction opposite to that of the torticollis. What is typical of CMT is the flattening of the skull in the parieto-occipital region [5,8]. This deformity is caused by pressure on the lambdoid suture at the end of pregnancy (Figure 4). Longer-lasting torticollis is characterised by changes that occur outside the cervical spine region. As the child grows, flattening of the parieto-occipital region and hair loss become noticeable, indicating that the deformity has persisted for a long time. Facial asymmetry also occurs over time and becomes more pronounced as children age. The face is shortened and diminished in the longitudinal dimension along the forehead—chin axis on the torticollis side. In the transverse dimension, the face is widened along the nose—ear axis.

The contracted muscle is firm and tense in infants over five months of age, with either a narrow or wide fibre course. Occasionally, muscle fibres split peripherally into two tense bands. With aging, deformities within the head and spine regions begin to exacerbate. During this period, a child can move towards the torticollis side, which is not observed in a healthy newborn; however, this movement is usually restricted. An asymmetrical head position can lead to an abnormal bite and impaired visual development [22]. The involuntary misalignment of the eyeballs results in the incorrect perception and processing of images [5]. The spine deforms as the cervical vertebrae adapt to flexion and become wedge-shaped [23].

### 1.4. Radiological Image

The correct diagnosis of torticollis requires further examination. In cases of typical CMT, comparative ultrasound scans of both SCMs are performed. An ultrasound examination may reveal changes in the structure of the SCM in the form of muscle thickening, fibrosis, or haematoma. This image confirms the diagnosis of the muscular origin of torticollis [19,24]. A diagnosis requires classical cervical spine radiology in the posteroanterior and lateral projections when clinical imaging indicates abnormalities in the skeletal system. In the case of a discordance between the clinical symptoms and the additional examination results, an extension of the diagnostics should include a computed tomography scan or magnetic resonance imaging.

### 1.5. Treatment

The treatment of torticollis aims to restore the normal function of the SCM by means of the following: restoring the anatomical length of the SCM muscle, restoring normal mobility of the cervical spine, achieving the muscle’s flexibility to enable normal movements of the head, and strengthening the weak groups of muscles. Most commonly, treatment begins with the implementation of nonsurgical methods. Therapy aims to achieve flexibility, correct the length of the SCM muscle, and eliminate its thickening. An early and proper treatment results in restoring symmetry within the head and trunk regions and attaining the psychomotor development corresponding to the calendar age. Surgical treatment is performed when conservative treatment fails. The essence of surgical treatment is to remove fibrosis and lengthen the SCM.

#### 1.5.1. Conservative Treatment

Proper physiotherapeutic management offers the possibility of significant improvement, provided that the therapy is implemented in the initial months of the infant’s life [7,25]. The most common management options include passive stretching of the shortened muscle and massaging the SCM [25]. These procedures aim to lengthen the affected muscles, achieve flexibility, and eliminate muscle thickening. Strengthening the contralateral side is equally important, as the opposite muscle is elongated and weakened. The therapeutic goal is to equalise the length and tension of both SCM muscles. Parental education plays a vital role in the treatment. Parents learn to care for their child properly, to position, stimulate, carry, change, and feed the child. Correctional positioning of the newborn in a prone position, with head rotation towards the torticollis side, is applied. This position is intended to correct SCM deformities. Parents are also taught to provide massages under home conditions. The physiotherapeutic methods most commonly applied in CMT treatment include the NDT-Bobath and Vaclav Vojta’s methods [17].

#### 1.5.2. Botulinum Toxin Type A (BTX-A)

In CMT, botulinum toxin type A (BTX-A) injections are a safe and effective treatment with few serious adverse reactions [26]. A meta-analysis conducted by Qiu et al. showed that the overall effective rate of botulinum toxin for congenital muscular torticollis was 84% [26]. An indication that BTX-A should be used is the presence of a firm, shortened muscle that has failed to be corrected using the abovementioned methods only [27]. A retrospective review of 39 children with resistant congenital muscular torticollis, which analysed the group in terms of head tilt correction, the range of neck motion improvement, the need for repeated BTX injections, and caregiver satisfaction, offered notable results [27]. The role of botulinum toxin in the treatment of torticollis may be more important than easing tension and lengthening the muscle. Studies by Jiang et al. pointed to a potential role for BTX in remodelling the fibrotic muscle through regulating fibroblast and inhibiting myofibroblast differentiation [28]. The data presented above should prompt clinicians to consider the early administration of BTX in cases of advanced deformities and extensive fibrotic lesions, which indicate a long-term treatment process.

#### 1.5.3. Surgical Treatment 

Surgical treatment is performed when conservative methods fail to improve the condition. Indications that surgery is needed include a marked shortening of the SCM muscle, persistent muscle fibrosis, persistent head and facial asymmetry, and rotation or lateral flexion in the cervical spine region restricted by >15° [27,29]. Most authors believe that surgery should be performed around one year of age. Surgical intervention is not offered earlier, as most changes (approximately 90%) tend to regress after nonsurgical treatment. However, if surgery is postponed, asymmetry in the head position can lead to a sideways spine curvature [30]. Moreover, some authors recommend performing surgery before one year of age to prevent skull deformities [31,32]. However, according to Shim and Jang, early surgery can create problems in postoperative wound management, because of the easier formation of haematomas and the increased prevalence of infection in younger children. Certain authors suggest that ages from 1 to 4 years are optimal for surgical intervention [33].

The most commonly performed surgeries are tenotomies and myotomies of the SCM, whereas lengthening the scarred muscles is less common (Figure 5). Tenotomies are performed as unipolar ones or, less commonly, by the subcutaneous method or with total exposure of tendon attachments. Unipolar tenotomy can be either upper or lower. Lower tenotomy is most widely applied, and it involves the release of the SCM muscle attachments on the clavicle and sternum. Upper tenotomy involves cutting the muscle near its attachment to the mastoid process. Bipolar tenotomy involves cutting the muscle near its attachment to the clavicle and sternum and near its attachment to the mastoid process. The bipolar method is used when the deformity is severe and cannot be corrected using the unipolar method. An incision is made on the wrinkled skin above the clavicle. Incisions in the clavicle or in its close proximity tend to widen and become unsightly. It is recommended that approximately 1 cm of the tendon be removed to prevent the recurrence of deformity. A transverse myotomy is performed at the mid-length of the SCM. Muscle lengthening is carried out via the “Z” plasty technique [33,34]. Early postoperative complications include wound damage, haematoma formation, and superficial wound infections. However, late complications include scar disfigurement, scar tethering, and the loss of SCM muscle contour [34]. The medical literature describes several endoscopic surgical techniques used to release the SCM. The advantages of endoscopy include its low invasiveness and minimal scarring [35]. Tang et al. recommends the endoscopic release of the SCM muscle in infants and children as young as six months of age [36]. Opinions in the literature on physiotherapeutic management after surgical treatment are divided. Angoules believes that, because recurrences occur in 1.2% of patients, an intensive physiotherapy program (including manual stretching of the SCM) is required [37]. Some authors recommend using hypercorrective head positioning after surgery, fixed with a plaster dressing, whereas others recommend a semi-rigid orthotic or cervical collar. Most authors agree with the guidance offered by physiotherapists to remodel the cortical map’s movement patterns after wound healing.

The aim of this study was to collect data on infants referred to paediatric rehabilitation and to identify the risk factors associated with CMT in this group of patients, as well as to assess demographic and clinical characteristics concerning risk factors.

## 2. Materials and Methods

This study was based on a retrospective analysis of children with CMT treated at the rehabilitation department of the Professor Jan Bogdanowicz Children’s Hospital in Warsaw, Poland, from 2000 to 2018. 

### 2.1. Study Group

The target population for this retrospective study consisted of 111 infants aged 0 to 5 months born in Poland and diagnosed with and receiving treatment due to CMT. The study group included 50 girls (46%) and 61 boys (54%). The group characteristics are presented in Table 1.

Each infant’s neck was examined in the supine position following a rotation of the head to the opposite side. Longitudinal views of the SCM muscle were obtained using conventional B-mode imaging. 

In this study, all children presented an increased thickness of the SCM. Abnormal echogenicity of the SCM was diagnosed in 66 patients (76%), including 31 female (36%) and 35 male (40%) infants; these changes were not noted in 21 patients (24%). The latter group comprised 7 girls (8%) and 14 boys (16%). Of the 66 children with abnormal echogenicity, 38 infants had left-sided torticollis (58%), and 28 infants had right-sided torticollis (42%). The decreased mobility and passive range of motion (PROM) restrictions in the cervical region of the spine were observed in 78 CMT infants (70%). In the study group, complications during delivery occurred in 35 children (33%); in 21 cases, the birth occurred via a caesarean section (20%), 10 births occurred via assisted vaginal delivery (9%), breech delivery was noted in 3 cases (3%), and a bigeminal pregnancy was reported in 1 patient (1%). 

### 2.2. The Inclusion Criteria for the Analysis Were as Follows

The inclusion criteria were as follows: (1) the calendar age of 0 to 5 months; (2) the diagnosis of torticollis of muscular origin with a palpable neck mass and a thickness of the involved sternocleidomastoid (SCM) muscle greater than 2 mm in comparison with the contralateral side (bone lesions were excluded); (3) increased echogenicity of the SCM in an ultrasonographic examination; (4) no trauma in the shoulder region—a clavicle fracture did not occur; no medical contraindications to participate in the research study; and informed and voluntary consent provided by the participants’ parents.

### 2.3. The Following Relationships Were Analysed

The relationships between the side of the CMT location and the type of delivery (cc vs. vaginal), between body weight at birth and the side of the CMT location, and between the extent of SCM muscle thickening and the type of delivery (cc vs. vaginal) were analysed. Whether or not the incidence of CMT depends on the delivery order was also determined.

### 2.4. Statistical Analysis

For statistical calculations, tests were applied depending on the distribution and variance homogeneity of the variables. The Mann–Whitney non-parametric U test was used when the distribution was not normal. 

### 2.5. Ethics 

All the procedures performed in this study involving human participants conformed to the ethical guidelines of the 2013 Declaration of Helsinki, as reflected in the prior approval from the institution’s human research committee. The protocol of the study was approved by the Institutional Review Board (IRB) and ethics committee (Military Medical Institute, Warsaw, Poland; number 124/WIM/2018). The written informed parental consent was waived due to the characteristics of the retrospective study. The approval was obtained from an institutional research ethics committee before recruitment, and the subjects received no payment for participating in this study. The experiment was conducted with the understanding of each patient’s parent. The photographs and epidemiological files of the children used in this article are for scientific and educational purposes, and informed consent has been obtained from the parents.

## 3. Results

### 3.1. Relationship between the Side of CMT Location and the Type of Delivery

In 53 (91%) cases, left-sided torticollis occurred in children born via vaginal delivery, and only 5 cases (9%) were delivered via cc. However, right-sided torticollis occurred in 37 children (69%) born via vaginal delivery, and 16 (31%) were born via cc. 

### 3.2. Relationship between the Birth Body Weight and the Location of CMT

The mean body weight at birth in children with left-sided CMT was 3674 g, whereas for those with right-sided CMT, it was 3274 g (Table 2). 

### 3.3. Relationship between the Extent of the SCM Muscle Thickening and the Type of Delivery (cc vs. Vaginal)

The thickness of both SCM muscles was analysed using ultrasonography. The difference in the thickness of both (right/left) SCM muscles was greater in children born via vaginal delivery and was 6.07 mm, on average. On average, the difference in the thickness of both SCM muscles in children born via a caesarean section was 4.14 mm (Table 3). The average width of the left SCM in left-sided torticollis in children born via vaginal delivery was 13.99 mm, whereas that of the right SCM in right-sided torticollis was 8.3 mm. Of the 57 cases of left-sided torticollis in the entire study group, an abnormal SCM muscle structure was noted in 38 patients (67%). Muscle structure abnormalities were reported in 28 (42%) of 54 cases of right-sided torticollis. In children with abnormal SCM structures, the difference in the thickness of both muscles was greater than that in children with normal SCM structures (6.15 vs. 3.79).

### 3.4. The Incidence of CMT Depending on the First or Subsequent Mode of Birth

Seventy-six percent (76%) of the paediatric patients diagnosed with CMT were the offspring of primipara mothers, including 41 female (37%) and 43 male infants (39%). Twenty-seven study participants (24%) were the offspring of multiparous women, including 11 girls (9%) and 16 boys (15%).

## 4. Discussion

### 4.1. Introduction

Congenital muscular torticollis (CMT) is a congenital condition that delays gross motor skills development in infants [5]. CMT is characterised by the shortening or excessive contraction of the sternocleidomastoid (SCM) muscles, which causes lateral flexion of the head to the side of torticollis, with rotation towards the opposite side and other accompanying asymmetries [38]. CMT can contribute to plagiocephaly, developmental dysplasia of the hip, brachial plexus trauma, and anomalies of the feet or lower limbs [8,31]. CMTs are most often located unilaterally and seldom bilaterally. Bilateral torticollis was not diagnosed in any of the 111 patients in this study. 

### 4.2. The Perinatal Risk Factors Associated with Congenital Muscular Torticollis—Sex Differences

It has been shown previously that CMT is more common in male than in female infants, which has been confirmed by many authors [10,39,40]. 

A study by Petronic et al. showed a higher percentage of CTM in male infants [40]. The current study also included more boys with CMT (61; 54%) than girls (51; 46%).

### 4.3. The Perinatal Risk Factors Associated with Congenital Muscular Torticollis—Infant’s Birthweight 

The analysis of body weight at birth revealed that male infants had a greater weight (3632 g) than females (3299 g), which may be the reason for the higher prevalence of CMT in male infants. Larger head circumference and greater body weight in boys may be risk factors.

### 4.4. The Perinatal Risk Factors Associated with Congenital Muscular Torticollis—Side of the CMT

The results obtained by Kim et al. demonstrated that CMT is more frequently associated with the left SCM muscle [41]. In this study, with a small difference, there were more cases of left-sided torticollis (*n* = 57; 51%) than right-sided torticollis (*n* = 54; 49%). Male infants more often had left-sided torticollis (58%), while in the group of female infants, left-sided torticollis occurred in 24 out of 57 cases (42%). This is consistent with studies showing that right-sided torticollis is more common in girls [40]. 

### 4.5. The Aetiology of Torticollis

The aetiology of CMT remains unclear. Stretching of the neck during difficult deliveries (often in large infants), combined with haemorrhages into the sternocleidomastoid muscle, leads to the increased incidence of CMT [42]. Some researchers consider inadequate blood supply to the SCM muscle, which may result from the problems during normal vascular development in the early stages, venostasis, or ischaemia due to the pressure on a blood vessel. Blood supply disturbances result in degenerative changes with the ultimate fibrosis and shortening of the muscle [11,43]. Most of the ischaemic theory supporters unanimously claim that ischaemia of the SCM muscle is a consequence of the non-physiological position of the foetus resulting from intrauterine space restriction [43]. 

It should also be noted that the shortened SCM muscle can become twisted during labour, resulting in a mechanical trauma and the formation of haematoma and secondary scarring. 

### 4.6. The SCM Characteristics and Cervical Range of Motion (CROM)

All children in the study group had the thickening of the SCM muscle, with as many as 78 children (70%) presenting the range of motion (ROM) restricted in the cervical region of the spine. In the remaining 34 cases (30%), there was no restriction in mobility in the cervical region of the spine. Therefore, this appears to support Tachdjian’s research, which confirmed that changes in the SCM muscle resulted in its shortening [44]. It is well known that SCM thickening and muscle alternations are associated with SCM tightness. 

Moreover, poor cervical spine rotation is associated with the decrease in thoracic rotation. The increased stiffness and asymmetry of the cervical spine could create congestion of the vascular and nervous structures located in the region of the cervical spine, impairing motor control and strength. 

This study showed that decreased mobility was associated with higher values of the difference in thickness between the ipsilateral and contralateral SCM muscles as well as the ipsilateral/contralateral ratio. It was shown previously that mobility restrictions of the cervical spine are considered one of the main predisposing factors for positional plagiocephaly in infants aged 4 month on average [45]. Plagiocephaly was present in 40.7% of babies with torticollis (*p* = 0.005) and joint limitation in 44.9% of babies (*p* < 0.001) [18]. 

This appears to support Tachdjian, who proposed that changes in the SCM resulted in its shortening [44]. According to this, CMT results from intrauterine space restriction, and the ischaemic theory appears to be most supported. This finding may be demonstrated by the number of first-born children (*n* = 84, 76%) recorded in this study. 

### 4.7. The Characteristics of Delivery

Since the average body weight at birth of children with CMT born during a mother’s first delivery was 3464 g, it may be assumed that such children could have had less space in the womb of primiparous mothers. The neonatal positioning may be asymmetrical in primiparas, with the head tilted to one side, lateral rotation towards the opposite side, and the maxillary area tilted towards the shoulder. 

CMT is relatively often observed not only when the foetus is in the breech position, but also in children of primiparas [46]. The data obtained in this study confirmed the latter and revealed that the incidence of congenital muscular torticollis was higher in first-born children. Ho et al. reported an incidence of 53% in children whose mothers were primiparous, and there was a high occurrence of traumatic childbirth. It was usually identified in neonates by age 2 to 3 weeks and could persist until the age of 1 year. It is typically unilateral, but occasionally it can be bilateral. There is a visible, palpable swelling known as a sternomastoid tumour, which appeared in 50% of cases [47].

### 4.8. Body Weight at Birth and the Location of CMT

In the analysed material, a significant relationship was observed between the body weight at birth and the location of CMT (*p* = 0.0001). The mean body weight at birth in children with left-sided CMT was 3674 g, while in children with right-sided CMT, it was 3274 g. The analysis also demonstrated a significant relationship between birth weight and sex (*p* = 0.0012). In the study group, boys had a greater body weight at birth than girls. The mean body weights at birth were 3632 g for male infants and 3299 g for female infants. In the study group, there was a predominance of boys with left-sided torticollis, who were also more numerous in the more severe torticollis group. Therefore, it can be presumed that a more severe form of CMT most likely occurs in boys and most commonly in the left SCM. 

Other authors showed that body weight at birth was the only risk factor identified in patients with a contralateral clavicular fracture followed by congenital muscular torticollis [48]. Moreover, there was no significant difference in the treatment duration between CMT infants with or without a clavicular fracture.

Other research showed that the baby’s gender, mode of delivery and the foetal presentation during delivery did not reveal a statistically significant association with the presence of torticollis [18].

The latter is the reason why it could be interesting in future studies to perform an analysis based on the correlation between the severity of male and female infants with CMT. 

This study also revealed that the thickness of the SCM muscle was significantly associated with an infant’s body weight. This is of the utmost importance, because children with a thicker SCM require a longer duration of therapeutic interventions, hence the thickness of the SCM may be a prognostic factor for CMT treatment [49].

### 4.9. The Association between the Thickness of SCM Muscle and the Mode of Delivery

In this study, a statistically significant difference was observed between the extent of SCM thickening and the mode of delivery (*p* = 0.02). A greater difference between the thickness of both SCM muscles was noted in children born via vaginal delivery (average: 6.07 mm) than in those born via a caesarean section (average: 4.14 mm). Of the 90 births delivered via vaginal labour in the study group, left-sided torticollis occurred in 52 neonates. Out of 21 births delivered via a caesarean section, left-sided torticollis occurred in only five cases. Therefore, 52 (91%) of the 57 patients with left-sided torticollis were delivered via vaginal labour. As shown on ultrasound scans, an abnormal SCM structure occurred in 66 patients in the study group, with left-sided torticollis noted in 38 patients (58%). The abnormal thickness of the left SCM was 13.99 mm while that of the right one was 8.3 mm, on average, in children born via vaginal delivery. The greater thickness of the abnormal left SCM demonstrated in the current study appears to be consistent with reports on the formation of micro-traumas during labour conditional on SCM muscle shortening [50,51]. Therefore, children delivered via vaginal labour in the study group had left-sided torticollis with greater muscle broadening than those with right-sided torticollis. The current study showed that greater thickening of the left SCM could have developed during vaginal labour. 

## 5. Conclusions

Theories of torticollis development pathophysiology should be deepened and systematised. Conducting observational and prospective studies on larger groups of patients is needed for the identification of the risk and prognostic factors. 

The data obtained in this study revealed that CMT is less common in female (*n* = 51; 46%) compared to male (*n* = 61; 54%) infants, in which a greater birth weight was reported (*p* < 0.05). The average body weight at birth was 3632 g for male infants vs. 3299 g for female infants. The birth weight and various characteristics of mothers appeared to be clinical predictors of CMT in infants; thus, more studies are needed on whether any screening and/or professional health intervention should be recommended to detect the risk of CMT. 

Most children with CMT were primiparous (76% vs. 24%). More often, children born via vaginal delivery had left-sided torticollis with a more significant broadening of the SCM muscle, as shown on ultrasound scans, than in right-sided torticollis. 

## Figures and Tables

**Figure 1 healthcare-12-00013-f001:**
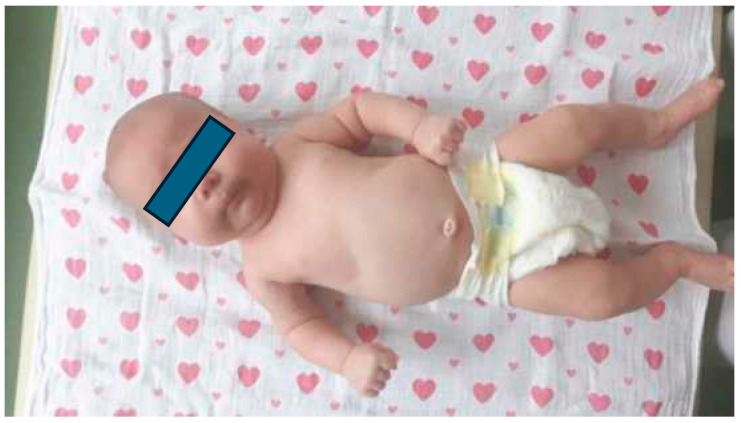
Characteristic asymmetrical head and trunk position in a “C” shape and oblique pelvic position during supination. Left-sided CMT.

**Figure 2 healthcare-12-00013-f002:**
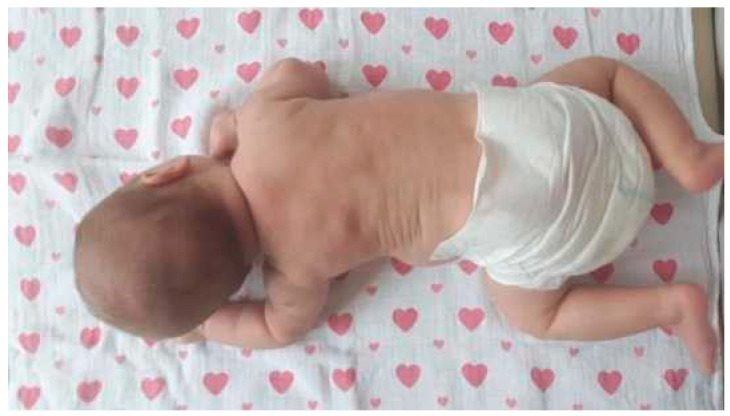
Characteristic asymmetrical head and trunk position in a “C” shape and oblique pelvic position in pronation. Left-sided CMT.

**Figure 3 healthcare-12-00013-f003:**
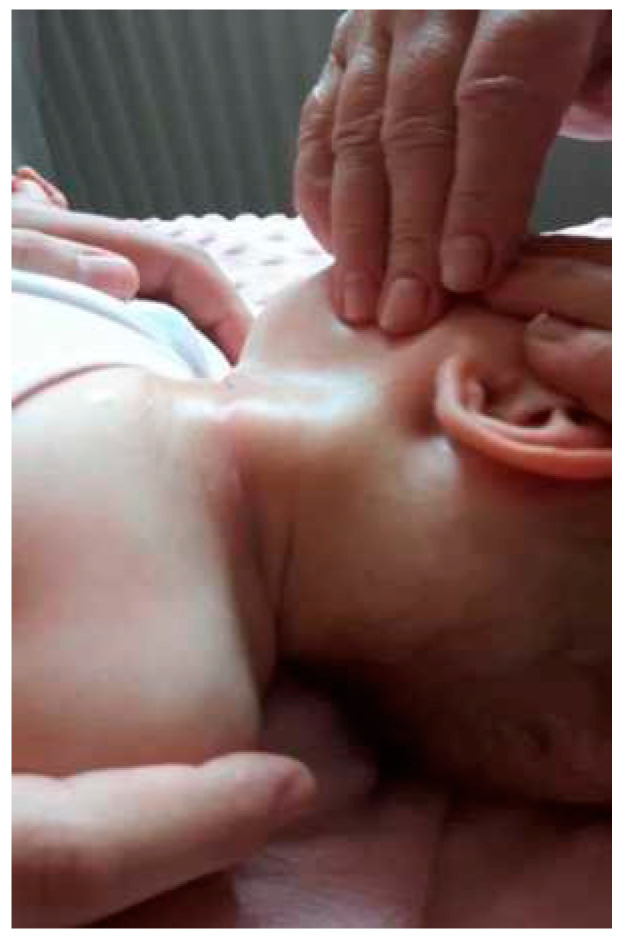
Noticeable thickening of the SCM muscle—left-sided CMT.

**Figure 4 healthcare-12-00013-f004:**
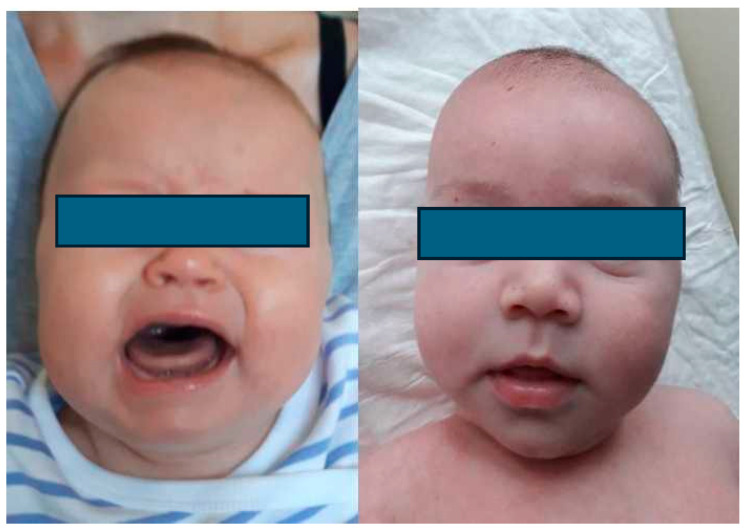
Head and facial asymmetry in an infant with left-sided CMT, flattening the head in the parietal region, shortening the face on the torticollis side in the longitudinal dimension along the chin–forehead axis, narrowing the eye fissure, lowering the forehead on the torticollis, and enlarging the cheek on the unaffected side.

**Figure 5 healthcare-12-00013-f005:**
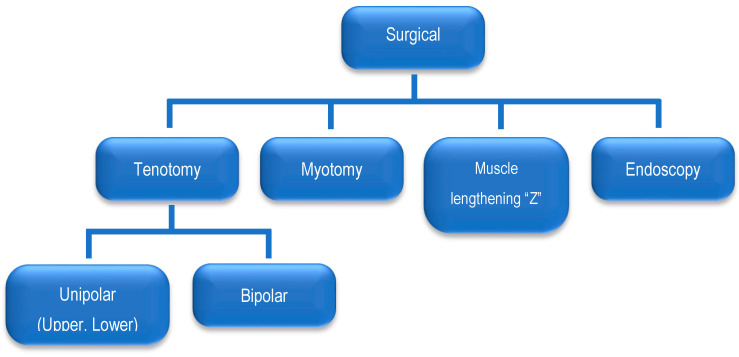
Types of surgery used to treat children with CMT.

**Table 1 healthcare-12-00013-t001:** Study group characteristics.

	N (%)	Left-Sided CMT	Right-Sided CMT	Mean Birth Weight	Birth		Term of Birth	
					cc	Vaginal	Pre	on Time
Male	61 (54%)	33	28	3632	10	51	1	60
Female	50 (46%)	24	26	3299	11	39	5	45
Total	111 (100%)	57	54		21	90	6	105

N—number (percentage); CMT—congenital muscular torticollis; cc—caesarean section.

**Table 2 healthcare-12-00013-t002:** Relationship between the side of the CMT location, mean body weight at birth, and the type of delivery.

	Vaginal Delivery	Caesarian Section	Mean Body Weight at Birth
L-sided torticollis	91%	9%	3674
R-sided torticollis	69%	31%	3274

**Table 3 healthcare-12-00013-t003:** A comparison of the sternocleidomastoid (SCM) muscle thickness in the ultrasound scan with the type of delivery, the number of births, sex, and the location of torticollis shown in millimetres.

	Thickness of R-SCM [mm]	Thickness of L-SCM [mm]	Difference in Thickness [mm]
Vaginal childbirth	8.30	13.99	6.07
Caesarean section	9.39	17.72	4.14
First birth	8.58	14.94	5.47
Next birth	8.30	14.00	6.35
Boys	8.40	14.56	5.86
Girls	8.67	14.92	5.44
L-CMT	4.93	10.69	5.76
R-CMT	11.22	5.61	5.61

Note: L—left; R—right.

## Data Availability

The project did not require registration. The datasets may be shared via correspondence with the authors upon a reasonable request.

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
