# Peer review of "Congenital Muscular Torticollis—Current Understanding and Perinatal Risk Factors: A Retrospective Analysis"

_healthcare, 2023, doi:10.3390/healthcare12010013_

Round 1
Reviewer 1 Report
Comments and Suggestions for Authors
Many of the topics of this paper have been addressed previously many times. The authors need to point out the unique contribution that is being made by this study and manuscript.
Throughout the manuscript, conservative therapy is said to "offer the chance of a complete cure"
Most of the references are from 2017 and much earlier. This becomes an issue when the authors purport to be presenting current state of particular interventions. The most glaring case is when the authors speak about the use of Botulinum toxin A for infants and cite literature from 2005-2007 that suggests that the safety of use of Botox for infants needs further study.
One significant omission from the references and the manuscript are the 2 clinical practice guidelines from the Pediatric Academy of the American Physical Therapy Association - published in 2013 & updated in 2018. It provides a synopsis of some of the information the authors cited but also summarizes much more with an assessment of the strength of each contribution.
The state of practice that is described is also from 2018 and prior so it is unclear how well that relates to the state of management 5 years later in 2023.
Lines 110 - 111: What is meant by the sentence, "In CMT, the range of cervical spinal mobility is typically normal"? That seems to negate the definition the authors gave CMT initially.
There is no mention of figure 5 - it seems out of place. Also, what is the resource or reference for use of a cervical collar in children with torticollis? Is that used in place of the TOT collar? Are there safety concerns with keeping a collar around a young infant's neck?
Article 22 by Lee (2013) was cited as stating that the vertebrae wedge in response to torticollis? The abstract made no mention of wedged vertebrae which would seem to have been an unusual and interesting finding. Is this accurate?
In the treatment section, PT was described as doing several things but strengthening the long, weak opposing muscle was not one of them. Seems like a rather big omission.
lines 196-197 - this sentence is confusing as it was previously stated that conservative therapy offers the chance of a full cure
lines 271-274 - the statement about informed consent indicates that it was waived due to the retrospective nature of the study yet the last sentence in that paragraph says that the study was conducted with the understanding of each patient's parents? This seems inconsistent.
Comments on the Quality of English LanguageSeveral choices of wording are awkward and sometimes inaccurate. Examples:
Introduction line 1: "...forced and asymmetrical head positioning..." - perhaps involuntary is what is meant?
Line 51: "Some authors prove that torticollis coexists with hip dysplasia" - perhaps "maintain" rather than "prove"
line 53-54: "...contributes to the rapid regression of torticollis symptoms" "rapid resolution" might be a better choice
line 126: "after age..." should be replaced by "With aging, ..." or "as the infant ages"
line 155 "To date no improvement has been recognized as the most effective method." This sentence needs modification
line 163 "forced passive stretching" seems much too strong and potentially harmful
line 167 "pronation" should read prone
Author Response
11 /05/2023
Response to the comments from the Reviewers
We would like to thank the reviewers for carefully reading our manuscript titled Congenital Muscular Torticollis – Current Understanding and Perinatal Risk Factors. A Retrospective Analysis and the time spent reviewing it.
The manuscript has been checked and revised by the authors, and the mistakes have been changed according to the reviewer's comments. All corrections have been made (and highlighted in red colour) in the attached file of the manuscript. We are thankful for all comments. We have done our best to answer them precisely and improve the manuscript.
Review 1
- Many of the topics of this paper have been addressed previously many times. The authors need to point out the unique contribution that is being made by this study and manuscript.
Response
The described and analyzed new group of patients broadens the base of publications, which can provide material for meta-analysis in the future. On this basis of the related risk factors, risk groups should be recommended for diagnosis and, once CMT is identified, for treatment—level of evidence III. Reliable, valid, and clinically helpful screening and risk assessment procedures should be documented and based on as large a research group as possible—quality of evidence: V; Strength of recommendation.
- Throughout the manuscript, conservative therapy is said to "offer the chance of a complete cure."
Response
This finding is based on data from the literature: studies have found that the prognosis for complete resolution of CMT was in children treated conservatively before three months of age. Five factors are associated with complete or more complete resolution of symptoms: (1) participation in the PT intervention, (2) younger age at the start of treatment, (3) reduction in the PROM cervical rotation discrepancy between sites, (4) reduction in the SCM muscle thickness discrepancy between sites, and (5) the caregiver's ability to frequently implement an active positioning and passive stretching program at home.
Kaplan, S., L., Coulter, C., Sargent, B. Physical therapy management of congenital muscular torticollis: a 2018 evidence-based clinical practice guideline from the APTA Academy of Pediatric Physical Therapy. Pediatr Phys Ther. 2018; 30:240–290.
Carenzio, G., Carlisi, E., Morani, I., Tinelli, C., Barak, M., Bejor, M., Toffola, E. Early rehabilitation treatment in newborns with congenital muscular torticollis. European Journal of Physical and Rehabilitation Medicine. 2015; 51 5, 539-545.
- Most of the references are from 2017 and much earlier. This becomes an issue when the authors purport to present the current state of particular interventions. The most glaring case is when the authors speak about the use of Botulinum toxin A for infants and cite literature from 2005 -2007 that suggests that the safety of the use of Botox for infants needs further study.
Response
Thank you for this comment; we rewrote this part of the article and supported it with a to-date article.
- Qiu, X., Cui, Z., Tang, G., Deng, H., Xiong, Z., Han, S., & Tang, S. The Effectiveness and Safety of Botulinum Toxin Injections for the Treatment of Congenital Muscular Torticollis. Journal of Craniofacial Surgery. 2020; 31(8): 2160 - 2166,
- Limpaphayom, N., Kohan, E., Huser, A., Michalska-Flynn, M., Stewart, S., Dobbs, M. Use of Combined Botulinum Toxin and Physical Therapy for Treatment-Resistant Congenital Muscular Torticollis. Journal of Pediatric Orthopaedics, 2019; 39: 343–348.
- Jiang, B., Zu, W., Xu, J., Xiong, Z., Zhang, Y., Gao, S., Ge, S., & Zhang, L. (2018). Botulinum toxin type A relieves sternocleidomastoid muscle fibrosis in congenital muscular torticollis. International Journal of biological macromolecules. 2018;112: 1014-1020.
- One significant omission from the references and the manuscript is the 2 clinical practice guidelines from the Pediatric Academy of the American Physical Therapy Association - published in 2013 & updated in 2018. It provides a synopsis of some of the information the authors cited but also summarizes much more with an assessment of the strength of each contribution.
Response
Thank you for this comment; we cited these guidelines in the text.
- Kaplan, S., L., Coulter, C., Sargent, B. Physical therapy management of congenital muscular torticollis: a 2018 evidence-based clinical practice guideline from the APTA Academy of Pediatric Physical Therapy. Pediatr Phys Ther. 2018;30:240–290.
- Lines 110 - 111: What is meant by the sentence, "In CMT, the range of cervical spinal mobility is typically normal"? That seems to negate the definition the authors gave CMT initially.
Response
The cause of torticollis is structural changes in the SCM muscle. The structure of the cervical
spine, especially the shape of the vertebrae and the congruence of the joints, remains
undisturbed, which is responsible for the normal range of passive mobility.
6. There is no mention of Figure 5 - it seems out of place. Also, what is the resource or reference
for the use of a cervical collar in children with torticollis? Is that used in place of the TOT
collar? Are there safety concerns with keeping a collar around a young infant's neck?
Response
Due to unclear scientific consensus, we removed the pictures and related sentences in the text.
7. Article 22 by Lee (2013) was cited as stating that the vertebrae wedge in response to
torticollis? The abstract made no mention of wedged vertebrae, which would seem to have
been an unusual and interesting finding. Is this accurate?
Response
Thank you for this comment. We cited more suitable references:
22. Hussein, M., Yun, I., Lee, D., Park, H., & Oock, K., 2018. Cervical Spine Dysmorphism in Congenital
Muscular Torticollis. Journal of Craniofacial Surgery, 29, pp. 925–929.
8. In the treatment section, PT was described as doing several things, but strengthening the long,
weak opposing muscle was not one of them. Seems like a rather big omission.
Response
We added the important role of physio treatment – lines 165-167.
9. lines 196-197 - this sentence is confusing as it was previously stated that conservative therapy
offers the chance of a full cure
Response
Conservative treatment early and properly undertaken is characterized by high efficiency.
However, there are cases of persistent or untreated torticollis, and then surgery should be
considered.
10. lines 271-274 - the statement about informed consent indicates that it was waived due to the
retrospective nature of the study yet the last sentence in that paragraph says that the study was
conducted with the understanding of each patient's parents? This seems inconsistent.
Response
We agree and corrected this part of the text.
11. Comments on the Quality of English Language
Response
All linguistic remarks have been applied, and a professional translator has read the entire text.

Reviewer 2 Report
Comments and Suggestions for Authors
I am very pleased to have the opportunity to review this manuscript. This manuscript is entitled "Congenital Muscular Torticollis - A Retrospective Analysis", a report of a retrospective analysis of perinatal risk factors for congenital muscular torticollis.
I have read the manuscript and would like to make the following comments.
1.(Materials and Methods)
Are the CMT children included in this analysis the entire number treated between 2000 and 2018? If any cases were excluded, they should be described in detail. Also, please describe in detail the diagnostic algorithm for myasthenic cervical dystonia, whether all cases were diagnosed at this institution or whether there were cases referred from other hospitals. 2.
2.(Discussion)
After line 340, there is a description comparing birth weight, sex, and site of onset, but this is not a discussion, and should be discussed in the results section. Since it is stated that there was a significant difference in birth weight between boys and girls, it is necessary to correct for weight as a confounding factor and compare the data for other factors (left-right difference, severity of disease).
In order to facilitate the reader's understanding of these results, it is recommended that figures be inserted as appropriate.
Based on the results obtained from the above analysis, we would appreciate it if you could write a discussion section.
Author Response
11 /05/2023
Response to the comments from the Reviewers
We would like to thank the reviewers for carefully reading our manuscript titled Congenital Muscular Torticollis – Current Understanding and Perinatal Risk Factors. A Retrospective Analysis and the time spent reviewing it.
The manuscript has been checked and revised by the authors, and the mistakes have been changed according to the reviewer's comments. All corrections have been made (and highlighted in red colour) in the attached file of the manuscript. We are thankful for all comments. We have done our best to answer them precisely and improve the manuscript.
Review 2.
- Materials and Methods
Are the CMT children included in this analysis the entire number treated between 2000 and 2018? If any cases were excluded, they should be described in detail. Also, please describe in detail the diagnostic algorithm for myasthenic cervical dystonia, whether all cases were diagnosed at this institution, or whether there were cases referred from other hospitals. 2.
Response
Thank you for this question. All patients who met the inclusion criteria were included in the analysis. In retrospective studies, for example, the risk of losing a patient who fails to enroll is lower than in prospective studies, where the variability in group size is high. Cases of myasthenic dystonia were not analyzed. The inclusion criterion for the study was only diagnosed vertebrae of the fibromatosis colli type.
As suggested by the reviewer, the specific process of the participant's inclusion and exclusion were added, and the corrections have been put in the attached file of the manuscript:
The target population for this retrospective cohort study consisted of 162 infants born in Poland aged from 0 to 5 months old. A total number of 112 children infants were diagnosed with congenital muscular torticollis (CMT) – the increased thickness of SCM was observed in all the CMT patients.
The inclusion criteria included: 1) the calendar age from 0 to 5 months of age; 2) the diagnosis of torticollis of muscular origin with a palpable neck mass and the thickness of the involved sternocleidomastoid muscle (SCM) greater than 2 mm in comparison with contralateral side (bone lesions were excluded); 3) increased echogenicity of SCM in ultrasonographic examination; 4) no trauma in the shoulder region - clavicle fracture didn’t occur; no medical contraindications to participate in the research study, an informed and voluntary notification of participants’ parents.
The exclusion criteria included infants with neurodevelopmental disorders, no specific finding on ultrasonography; subjects who did undergo plain radiography of the cervical spine and/or clavicles were excluded – none of the children followed any trauma or fracture; the coexistence of other disabilities and diseases; congenital anomalies of the cervical spine; spasmodic, neurogenic and ocular torticollis, infants lost to follow-up; children with any inflammatory and/or infectious diseases of the neck; any medications that may affect the results of the analysis as well as those who were unable give valid consent. The exclusion criteria were also acute or chronic diseases such as heart problems, diabetes, asthma, inflammation, trauma, recent fractures of the bone, and recent surgery – in the last 6 months. Moreover, the participants who did not feel well at the time of recruitment were excluded from the analysis. Parents were instructed to maintain their children's accustomed dietary and physical activity habits throughout the study participation period.
- Discussion
After line 340, there is a description comparing birth weight, sex, and site of onset, but this is not a discussion and should be discussed in the results section. Since it is stated that there was a significant difference in birth weight between boys and girls, it is necessary to correct for weight as a confounding factor and compare the data for other factors (left-right difference, severity of disease). In order to facilitate the reader's understanding of these results, it is recommended that figures be inserted as appropriate. Based on the results obtained from the above analysis, we would appreciate it if you could write a discussion section.
Response
Variables describing birth weight, sex, and location of torticollis were shown in tables and described in the results section. In the discussion, we only compare our results with those of other researchers, which is why they are placed in the discussion section. We have checked the accuracy of the presented data once again.
Thank you for all your comments and suggestions.

Round 2
Reviewer 1 Report
Comments and Suggestions for Authors
This paper has some interesting data but is limited by interpretation using older literature and
The 2 categories of types of congenital muscular torticollis are unusual. Typically CMT is categorized as congenital and acquired and the congenital is further broken down into postural, muscular and SCM mass. As there is currently limited evidence to substantiate the causal mechanisms, the ultrasound evidence in this study would be of interest. However, the causal mechanisms presented in this paper are described as if they are factually accepted rather than simply a correlation. This is misleading.
In line 27 and throughout the paper, the terms "hip dislocations" and "foot deformities" are utilized to describe the 2 more common orthopedic issues in the newborn. It would be more accurate to state that hip dysplasia and equinovarus deformities are more common rather than hip dislocations and the entire category of foot deformities.
line 58 - omit "left"
line 174 should include strengthening - it is mentioned later but this overview only speaks to restoring normal mobility and flexibility.
lines 402-403 - assuming that CMT severity is worse in boys than girls is present seems like a big assumption based on lacking data. It seems that it would be a simple task to perform the correlation analysis to determine if that is the case.
Comments on the Quality of English Language
Some wording needs to be corrected to convey meaning accurately. Rather than "natural birth" I would suggest using the term "vaginal birth."
The authors say that prior researchers "proved" a relationship between two factors - instead it seems that the term "demonstrated" would be more accurate.
Use of "cure" for torticollis is better described as "resolution"
line 26 - "newborns and infants anomalies..." should be "newborn and infant"
Caesarian should be capitalized throughout
line 50 - not all CMT is "inborn"
lines 92-93 - this is confusing where the authors state that "abnormal head tilt may cause injury to the mother's womb due to muscle changes" and then goes on to say that the tilted head causes muscular injury - it is unclear if this is injury to the mother or the infant. It seems that the case should be made relative to the infant but this is a very confusing set of sentences.
line 129 - what is "olive-shaped" about the muscle? Or does this refer to the mass?
line 136 - flattening of the parieto-occipital region does not exclusively occur in utero - this part is misleading.
Line 149-150 - this sentence is confusing where it states that "the child can move toward the torticollis. which cannot be observed in newborns, however this movement is usually restricted."
line 303 - avoid use of contractions - use "did not" rather than "didn't"
line 369 and caption of a figure - for what is "CTM" the abbreviation?
line 389 - "...children born during their first delivery..." is misleading. All babies are born during their first delivery - it seems the authors mean during the mother's first delivery?
Author Response
17.11.2023
Response to the comments from the reviewer
We would like to thank the reviewer for careful reading of the manuscript and the time spend on reviewing the manuscript titled: Congenital Muscular Torticollis – Current Understanding and Perinatal Risk Factors. A Retrospective Analysis.
All manuscript has been checked and revised by authors and the mistakes have been changed according to the reviewer comments. All corrections have been made (and highlighted by red corrections) in the attached file of the manuscript.
We are thankful for all comments. We have done our best to answer precisely to them and improved the manuscript.
We are really sure that after revision the manuscript precisely addresses the issues raised in this study.
Sincerely,
Anna Malwina Kamelska-Sadowska
Submission Date
………………
Date of this review
……………………
COMMENTS FROM THE REVIEWERS:
OPEN REVIEW 3
- This paper has some interesting data but is limited by interpretation using older literature
Response:
As suggested by the reviewer, the novelty and the specific reasons why this study was valid has been added and all the discussion section has been rearranged and changed - the corrections have been put in the attached file of the manuscript. All literature data has been changed for more up-to-date e.g.
- Davids, J., R., Wenger, D., R., Mubarak, S., J. Congenital muscular torticollis: a sequela of intrauterine or perinatal compartment syndrome. J Pediatr Orthop. 1993; 13(2):141-147.
Has been changed for:
- 1. Gundrathi, J., Cunha, B., Mendez, M.D. Congenital Torticollis. [Updated 2023 Jan 31]. In: StatPearls [Internet]. Treasure Island (FL): StatPearls Publishing; 2023 Jan-. Available from: https://www.ncbi.nlm.nih.gov/books/NBK549778/
- Sanerkin, N., G. Birth injury to the sternomastoid muscle. J Bone Joint Surg Br. 1966; 48(3): 441- 447.
Has been changed for:
- Davids, J., R., Wenger, D., R., Mubarak, S., J. Congenital muscular torticollis: a sequela of intrauterine or perinatal compartment syndrome. J Pediatr Orthop. 1993; 13(2):141-147.
Line 89:
1.1. Aetiology
Usually, torticollis is not a diagnosis but rather a manifestation of a variety of underlying conditions. However, in general, torticollis is classified as either congenital (present at birth) or acquired (occurring later in infancy or childhood) disorder [1], and the congenital type is further classified into postural, muscular and SCM mass [9].
The literature most often cites the theory of muscle injury and remodeling during the foetal period, or the theory of an injury during delivery [1].
- Yu, C., C., Wong, F., H., Lo, L., J., Chen, Y., R. Craniofacial deformity in patients with uncorrected congenital muscular torticollis: an assessment from three-dimensional computed tomography imaging. Plast Reconstr Surg 2004;113(1): 24-33
Has been changed for:
- Kurniawan, A., Canintika, A. F. A rare case of 9 years congenital muscular torticollis treated with complete unipolar sternocleidomastoid release: A case report and literature review. Int J Surg Case Rep. 2022; 96: 107298.
Line 82:
Asymmetrical loads on the hip and pelvic joints can result in remote deformities. When left untreated, CMT may result in progressive limitation of head movement, which may end up in eye movement disorder, craniofacial asymmetry, malocclusion, visual defects, neck pain and compensatory asymmetrical spine curvature that worsens with age [8].
- Sanerkin, N., G. Birth injury to the sternomastoid muscle. J Bone Joint Surg Br. 1966; 48(3): 441-447.
Has been changed for:
- Wang, L., Zhang, L., Tang, Y., Qiu, L. The value of high-frequency and color Doppler ultrasonography in diagnosing congenital muscular torticollis. BMC Musculoskelet Disord. 2012, 26; 13: 209.
The newest data has also been added in the discussion section e.g.:
- Pastor-Pons, I., Lucha-López, M.O., Barrau-Lalmolda, M., Rodes-Pastor, I., Rodríguez-Fernández, Á.L., Hidalgo-García, C., Tricás-Moreno, J.M. Active Cervical Range of Motion in Babies with Positional Plagiocephaly: Analytical Cross-Sectional Study. Children (Basel). 2021; 6;8(12): 1146.
- Siegenthaler H.M. Unresolved Congenital Torticollis and Its Consequences: A Report of 2 Cases. J Chiropr Med. 2017; 16(3):257-261.
- Ho, B.C., Lee, E.H., Singh, K. Epidemiology, presentation and management of congenital muscular torticollis. Singapore Med J. 199 ; 40(11):675-9.
- Lee Z, Cho JY, Lee BJ, Kim JM, Park D. Body Weight at Birth: The Only Risk Factor Associated with Contralateral Clavicular Fracture in Patients with Congenital Muscular Torticollis. Sci Rep. 2019; 24;9(1): 13801.
- Han JD, Kim SH, Lee SJ, Park MC, Yim SY. The thickness of the sternocleidomastoid muscle as a prognostic factor for congenital muscular torticollis. Ann Rehabil Med. 2011 Jun;35(3):361-8.
- The 2 categories of types of congenital muscular torticollis are unusual. Typically CMT is categorized as congenital and acquired and the congenital is further broken down into postural, muscular and SCM mass.
Response:
We are really sure that the above comment of the reviewer is crucial, however from the clinical point of view the division of torticollis into type I and II were investigated.
The changed part of the manuscript now states:
“1.1. Aetiology
Usually, torticollis is not a diagnosis but rather a manifestation of a variety of underlying conditions. However, in general, torticollis is classified as either congenital (present at birth) or acquired (occurring later in infancy or childhood) disorder [1], and the congenital type is further classified into postural, muscular and SCM mass [9].
The literature most often cites the theory of muscle injury and remodeling during the foetal period, or the theory of an injury during delivery [1].
Concerning the clinical features the torticollis could be divided into two groups of mechanism.
CMT type I
Type I CMT is caused by fetal malposition [9].”
- As there is currently limited evidence to substantiate the causal mechanisms, the ultrasound evidence in this study would be of interest. However, the causal mechanisms presented in this paper are described as if they are factually accepted rather than simply a correlation. This is misleading.
Response:
We are really sure that the ultrasound evidence is indisputable. In the cases of typical CMT, comparative ultrasound scans of both SCMs were performed in this study.
The specific part concerning usg examination has been added in the manuscript:
Line 322:
“Each infant’s neck was examined in the supine position following rotation of the head to the opposite side. Longitudinal views of the SCM muscle were obtained from conventional B-mode imaging.”
- In line 27 and throughout the paper, the terms "hip dislocations" and "foot deformities" are utilized to describe the 2 more common orthopedic issues in the newborn. It would be more accurate to state that hip dysplasia and equinovarus deformities are more common rather than hip dislocations and the entire category of foot deformities.
Response:
As suggested by the reviewer, "hip dislocations" and "foot deformities" terms has been changed and the corrections have been put in the attached file of the manuscript:
Line 24, 70: “(…)Indications for surgery include marked shortening of the SCM, persistent fibrosis in the muscle, constant head and facial asymmetry, and rotation or lateral flexion in the cervical spine restricted by > 15°. Of all the newborn and infant anomalies, congenital torticollis is the third only to hip dysplasia and equinovarus deformities(…)”
- line 58 - omit "left"
Response:
As suggested by the reviewer, “left" has been omitted and the corrections have been put in the attached file of the manuscript:
Line 63: “Figure 1 shows the infant's characteristic tilt of the head, rotation to the opposite side of the affected muscle, and increased thickness of the affected muscle.”
- line 174 should include strengthening - it is mentioned later but this overview only speaks to restoring normal mobility and flexibility.
Response:
As suggested by the reviewer, “strengthening", as a following treatment has been added and the corrections have been put in the attached file of the manuscript:
Line 200: “(….) The treatment of torticollis aims to restore the normal function of the SCM by means of the following: restoring the anatomical length of the SCM muscle, restoring normal mobility of the cervical spine, achieving the muscle’s flexibility to enable normal movements of the head, and strengthening the weak groups of muscles.”
- lines 402-403 - assuming that CMT severity is worse in boys than girls is present seems like a big assumption based on lacking data. It seems that it would be a simple task to perform the correlation analysis to determine if that is the case.
Response:
As suggested by the reviewer, the discussion has been expanded by the following data concerning the CMT differences between male and female infants. The specific data confirmed that the CMT prevalence is higher in boys, especially because of the increased birthweight.
Line 419:
4.3. The perinatal risk factors associated with congenital muscular torticollis – infant’s birthweight
The analysis of body weight at birth revealed that male infants had a greater weight (3,632 g) than females (3,299 g), which may be the reason for the higher prevalence of CMT in male infants. Larger head circumference and greater body weight in boys may be risk factors.
A study by Petronic et al. showed a higher percentage of CTM in male infants [41]. The current study also included more boys with CMT (61; 54%) than girls (51; 46%).
The analysis of body weight at birth revealed that boys had a greater weight (3,632 g) than girls (3,299 g), which may be the reason for the higher prevalence of CMT in male infants. Larger head circumference and greater body weight in boys may be risk factors.”
Line 500:
“Other authors showed that body weight at birth was the only risk factor identified in patients with contralateral clavicular fracture followed by congenital muscular torticollis [49]. Moreover, there was no significant difference in the treatment duration between CMT infants with or without clavicular fracture.
Other research showed that the baby’s gender, mode of delivery and the foetal presentation during delivery did not reveal a statistically significant association with the presence of torticollis [18].
The latter is the reason why it could be interesting in future studies to perform analysis based on the correlation between the severity of male and female infants CMT.
This study also revealed that the thickness of SCM muscle was significantly associated with an infant’s body weight. It is of utmost importance because children with a thicker SCM require a longer duration of therapeutic interventions, hence the thickness of the SCM may be a prognostic factor for CMT treatment [50].”
Comments on the Quality of English Language
- Some wording needs to be corrected to convey meaning accurately. Rather than "natural birth" I would suggest using the term "vaginal birth."
Response:
As suggested by the reviewer, term “natural birth” has been changed for “vaginal childbirth” “vaginal birth” or “”vaginal delivery” and the corrections have been put in the attached file of the manuscript.
- The authors say that prior researchers "proved" a relationship between two factors - instead it seems that the term "demonstrated" would be more accurate.
Response:
As suggested by the reviewer, the term “proved” has been changed for “demonstrated”.
- Use of "cure" for torticollis is better described as "resolution"
Response:
As suggested by the reviewer, The term “cure” has been changed for “resolution”.
- line 26 - "newborns and infants anomalies..." should be "newborn and infant"
Response:
The corrections has been made, as suggested by the reviewer.
- Caesarian should be capitalized throughout
Response:
The term Caesarian has been changed for capitalized throughout the whole manuscript.
- line 50 - not all CMT is "inborn"
Response:
The definition of torticollis has been changed as suggested by reviewer and now states:
Line 55:
“Congenital muscular torticollis (CMT) is a condition commonly diagnosed at or soon after birth, characterized by involuntary and asymmetrical head position caused by the unilateral shortening of the sternocleidomastoid muscle (SCM), usually associated with fibrotic mass”
- lines 92-93 - this is confusing where the authors state that "abnormal head tilt may cause injury to the mother's womb due to muscle changes" and then goes on to say that the tilted head causes muscular injury - it is unclear if this is injury to the mother or the infant. It seems that the case should be made relative to the infant but this is a very confusing set of sentences.
Response:
We can’t agree more with the reviewer comment. The above mentioned sentence is quite confusion and misunderstood.
As suggested by the reviewer the part of this section has been changed for:
“Other authors claim that the injury to the SCM muscle may be due to excessive twisting of the head during its evacuation when the muscle fibers are torn, and a hematoma is formed, which subsequently undergoes resorption and fibrosis [13].”
- line 129 - what is "olive-shaped" about the muscle? Or does this refer to the mass?
Response:
As suggested by the reviewer the “olive-shaped” has been changed for “oval-shaped”.
- line 136 - flattening of the parieto-occipital region does not exclusively occur in utero - this part is misleading.
Response:
As suggested by the reviewer, the misunderstood part of the manuscript has been clarified (line 142).
- Line 149-150 - this sentence is confusing where it states that "the child can move toward the torticollis. which cannot be observed in newborns, however this movement is usually restricted."
Response:
Some of the parts of the manuscript could be misunderstood because of some language use mistakes. However, this has been changed after corrections made by the linguistic specialist.
The changed sentence is as follows:
Line 178: “During this period, a child can move toward the torticollis side, which is not observed in a healthy newborn;”
- line 303 - avoid use of contractions - use "did not" rather than "didn't"
Response:
All corrections has been made throughout the whole manuscript.
- line 369 and caption of a figure - for what is "CTM" the abbreviation?
Response:
Thank You for the spelling mistake notice. All corrections has been made.
- line 389 - "...children born during their first delivery..." is misleading. All babies are born during their first delivery - it seems the authors mean during the mother's first delivery?
Response:
All corrections has been made and the changed part of the manuscript states:
Line 491:
“According to this, CMT results from intrauterine space restriction, and the ischaemic theory appears to be most supported. This finding may be demonstrated by the number of children (n=84, 76%) born by first-time mothers recorded in this study.
4.7. The characteristics of delivery
Since the average body weight at birth of children with CMT born in the mother’s first delivery was 3,464 g, it may be assumed that such children could have had decreased space in the primipara’s womb.”

Reviewer 2 Report
Comments and Suggestions for Authors
I have confirmed the revised manuscript.
A table has been added to the results, which I think makes it easier for readers to understand.
However, the discussion part (from line 397) has not been modified. Again, the analysis becomes the results part, and any significant relationships need to be shown in the results part. Also, the confounding effect of body weight has not been corrected.
Please reconsider your manuscript.
Author Response
17.11.2023
Response to the comments from the reviewers
We would like to thank the reviewer for careful reading of the manuscript and the time spend on reviewing the manuscript titled: Congenital Muscular Torticollis – Current Understanding and Perinatal Risk Factors. A Retrospective Analysis.
All manuscript has been checked and revised by authors and the mistakes have been changed according to the reviewer comments. All corrections have been made (and highlighted by red corrections) in the attached file of the manuscript.
We are thankful for all comments. We have done our best to answer precisely to them and improved the manuscript.
We are really sure that after revision the manuscript precisely addresses the issues raised in this study.
Sincerely,
Anna Malwina Kamelska-Sadowska
Submission Date
………………
Date of this review
……………………
COMMENTS FROM THE REVIEWERS:
OPEN REVIEW 4
- A table has been added to the results, which I think makes it easier for readers to understand.
Response:
Thank You for the valuable comment.
- However, the discussion part (from line 397) has not been modified. Again, the analysis becomes the results part, and any significant relationships need to be shown in the results part. Also, the confounding effect of body weight has not been corrected.
Response:
As suggested by the reviewer, the discussion part has been rearranged and modified. The novelty and the specific reasons why this study was valid has been added, also more studies have been added and the corrections have been put in the attached file of the manuscript. All literature data has been changed for more up-to-date. The specific data concerning the body weight associations with CMT has been clarified.